# Study on Mechanical Behavior and Energy Mechanism of Sandstone under Chemical Corrosion

**DOI:** 10.3390/ma15041613

**Published:** 2022-02-21

**Authors:** Lei Chen, Baoxin Jia, Shuguang Zhang

**Affiliations:** 1Civil Engineering Institute, Liaoning Technical University, Fuxin 123000, China; cd9194198420@163.com; 2Guangxi Key Laboratory of Geotechnical Mechanics and Engineering, Guilin University of Technology, Guilin 541004, China; zhangll198212@163.com

**Keywords:** rock mechanics, sandstone, chemical corrosion, characteristic stress, energy damage

## Abstract

Chemical corrosion has a significant impact on the properties of rock materials. To study the mechanical behavior and energy mechanism of rock under chemical corrosion, this paper took the sandstone of Haitangshan tunnel in Fuxin as the research object, used a Na_2_SO_4_ solution to simulate different chemical environments, carried out a triaxial loading test on sandstone through the MTS815.02 test system, and analyzed the mechanical parameters and energy damage evolution law of sandstone under different chemical environments. The test results showed that the basic mechanical parameters (peak strength *σ*_pk_, peak strain *ε*_pk_, elastic modulus *E*, cohesion *c*, and internal friction angle *φ*) and characteristic stress parameters (closure stress *σ*_cc_, initiation stress *σ*_ci_, and dilatancy stress *σ*_cd_) of sandstone first increased and then decreased with the increase of pH in the Na_2_SO_4_ solution, Poisson’s ratio *µ* showed the opposite trend, and the extreme values of all parameters were taken when pH = 7. The influence degree of different pHs on the mechanical parameters of sandstone were as follows: strong acid environment (pH ≤ 4) > strong alkali environment (pH ≥ 10) > weak acid environment (4 ≤ pH < 6) > weak alkali environment (8 ≤ pH < 10) > neutral environment (6 < pH< 8). The total energy and elastic strain energy increased first and then decreased, and the dissipated energy was the opposite. The damage variable decreased first and then increased. With the increasing concentration of the Na_2_SO_4_ solution, all the above parameters changed monotonically. Based on the energy theory, the damage evolution equation considering the effect of the Na_2_SO_4_ concentration was established. Combined with the test data, the model was verified and the result was good. Under the action of Na_2_SO_4_ corrosion, Ca^2+^ in calcite and Fe^2+^ in hematite were dissolved and precipitated. With the gradual increase of Ca^2+^ and Fe^2+^ concentration, the damage variable increased gradually. The relationship between the two ion concentrations and the damage variable approximately satisfied a linear function.

## 1. Introduction

The surrounding rock of underground engineering contains a certain amount of water more or less [1,2]. As the most active factor in the geological environment, groundwater is essentially a complex chemical solution with different ionic components, concentrations, and pHs. The failure of surrounding rock in actual underground engineering is the result of the comprehensive action of excavation unloading and chemical corrosion, and it is an important factor affecting the stability of surrounding rock in underground engineering. Therefore, it is necessary to study the damage and deterioration mechanism of rock under chemical corrosion.

In recent years, some scholars have studied the physical and mechanical properties of rocks under chemical corrosion. Arindam et al. [3] conducted shear strength tests on jointed limestone under acid solution corrosion in view of the corrosion and degradation of jointed limestone under acid rain and analyzed the changes in shear strength parameters and failure mechanism before and after corrosion. Feng et al. [4] carried out a microscopic analysis of chemically corroded sandstone through CT scanning imaging technology and tested the strength through a compression test. Li et al. [5,6] analyzed the sandstone corroded by H_2_SO_4_ by means of NMR and SEM and revealed the metamorphic mechanism in an acidic environment. Gong et al. [7] studied the red sandstone with a single prefabricated crack after hydrochemical corrosion through a uniaxial compression test and analyzed the influence of different chemical solutions on the mechanical properties of sandstone. Lin et al. [8] analyzed the porosity of chemically corroded sandstone by NMR, analyzed the mechanical properties by uniaxial compression test, and established a damage constitutive model that can be used to describe chemically corroded sandstone. Han et al. [9,10,11] studied mode I fractured sandstone corroded by chemical solutions and freeze–thaw cycles through the three-point bending test, tensile test, and compression test, and analyzed the mechanical properties and failure mechanism under different test conditions. Yuan et al. [12] conducted uniaxial and triaxial compression tests on sandstone under dry-wet cycle and chemical corrosion conditions and analyzed the damage and failure mechanism from the perspective of thermodynamics and dynamics. Li et al. [13] conducted NMR tests on sandstone corroded by chemical solutions and after freeze–thaw cycles and analyzed the pore change law and damage deterioration mechanism. Qu et al. [14] studied the damage and deterioration mechanism of yellow sandstone under the coupling effect of chemical corrosion and freeze–thaw cycle through NMR, a freeze–thaw cycle test, chemical composition analysis test, and uniaxial compression test. Gutiérrez et al. [15] tested the chemically corroded sandstone by XRD, analyzed the ion distribution in the solution under an acidic environment, and revealed the corrosion mechanism under an acidic environment. Gao et al. [16] conducted NMR tests on red sandstone corroded by chemical solutions and freeze–thaw cycles, revealing the damage and deterioration mechanism. Xie et al. [17] analyzed limestone after chemical solution corrosion through compression tests and revealed the deformation and damage deterioration mechanism. Fang et al. [18] conducted a uniaxial compression test on yellow sandstone under chemical corrosion and a freeze–thaw cycle, and analyzed the mechanical properties and failure mechanism under different test conditions. Li et al. [19] conducted a triaxial test on sandstone corroded by chemical solutions and established a damage constitutive model considering the chemical corrosion effect. Qiao et al. [20] soaked sandstone in chemical solutions with different ion concentrations, analyzed the relationship between immersion time and ion concentration by SEM, and conducted a uniaxial compression test. It can be seen from the above that the existing research studies have carried out detailed analyses of the physical and chemical properties of sandstone in a chemically corrosive environment. The degradation mechanism was studied by different test methods, the variation law of mechanical parameters and chemical composition were analyzed, and a triaxial damage constitutive model was established considering the chemical corrosion effect, which provided an important theoretical basis for underground engineering construction. However, the above research lack comparison of mechanical parameters and characteristic stress changes under different pHs, which has an important impact on further understanding of mechanical properties of chemically corroded sandstone.

In physics, energy is the multiplication of force and displacement. Similarly, the rock will also deform under the action of external force, that is, the external force does work on the rock. Before the rock is destroyed, the energy generated by the work done by the external force is stored inside the rock. Once the energy storage limit of the rock is reached, the stored energy will be released instantly and the rock is destroyed. Energy mechanism analysis is to analyze the energy evolution law of rock by changing experimental conditions. As a new method in recent years, energy mechanism analysis has been gradually adopted by the majority of researchers. Zhang et al. [21] conducted numerical simulation research on sandstone under triaxial compression by PFC3D, analyzed the energy evolution law under loading mechanism with energy theory, and revealed the failure mechanism from the perspective of energy. Li et al. [22] studied the energy evolution mechanism and fatigue behavior of sandstone under cyclic load through a large number of tests and established the fatigue damage energy evolution equation. Zhou et al. [23] used the improved Hopkinson splitting compression bar to conduct an axial impact test on sandstone to analyze the energy damage evolution mechanism under dynamic disturbance. Luo et al. [24] carried out a numerical simulation of impact test on sandstone by PFC2D, and analyzed the energy evolution mechanism and crack propagation law under impact load. Yang [25] conducted triaxial test research on hollow cylinder sandstone, monitored the loading failure process based on X-ray and CT, and analyzed the crack evolution mechanism. Liu et al. [26] conducted a uniaxial compression test and stress relaxation test on prefabricated fractured granite and monitored the cracks in the rock through three-dimensional scanning technology. It can be seen from the above that the existing research has carried out a detailed study on the energy mechanism of the sandstone failure process, analyzed the energy evolution law through different test methods, established the energy damage evolution equation and triaxial constitutive model, and revealed the failure mechanism from the perspective of energy. It provided a new research method for surrounding rock deformation analysis of underground engineering construction. However, the above research lacks analysis of energy damage evolution mechanism and damage evolution equation of chemically corroded sandstone, which has an important impact on further understanding of the energy damage evolution mechanism.

To sum up, the existing results have conducted a detailed study on the physical properties and mechanical characteristics of rocks. However, there are few reports on the characteristic stress and energy damage evolution of rocks under chemical corrosion. Therefore, a Na_2_SO_4_ solution is used to simulate different chemical environments to carry out triaxial loading test on the chemically corroded sandstone of Haitangshan tunnel in Fuxin and analyze the mechanical properties and energy damage evolution law of sandstone under different pHs and concentrations, so as to provide a reliable theoretical basis for the analysis of sandstone failure mechanism under chemical corrosion.

## 2. Experimental Methodology

The triaxial loading tests of chemically corroded sandstone in this paper are completed on the MTS815.02 triaxial test system, as shown in Figure 1a. The sandstone is taken from Fuxin Haitangshan tunnel, with hard texture, uniform particle size, no obvious joints on the surface, gray in the natural state, dry density of 2.42–2.59 g/cm^3^, porosity of 0.48–0.56%, and particle size of about 0.01–0.42 mm (Figure 1b). In order to minimize the difference, all specimens are taken from the same complete rock block. According to ISRM, the complete rock block was cut, drilled, and ground, and we finally obtained 50 mm × 100 mm (diameter × height) standard cylindrical specimen. According to X-ray, the main chemical components of sandstone include quartz (SiO_2_), calcite (CaCO_3_), and hematite (Fe_2_O_3_), as well as traces of dolomite, chlorite, and other substances.

According to the actual buried depth of the tunnel and the measured in situ stress data, and in order to compare the parameter variation law of sandstone, triaxial loading test confining pressures are set as 5, 10, 15, and 20 MPa respectively. Na_2_SO_4_ solutions with different pH values and concentrations *α* are used to simulate the chemical corrosion environment of sandstone. See Table 1 for the specific test scheme. The test steps are as follows: (1) Firstly, place the prepared specimen in the drying box, set the temperature to 104 °C, and the drying time to 24 h. The dried specimen is placed in a drying dish and cooled to room temperature for standby. (2) All specimens are numbered and soaked in the solution in Table 1 according to the numbers for 7 d. (3) Take out all specimens and naturally dry for 48 h at room temperature for triaxial loading test.

## 3. Results

### 3.1. Analysis of Stress–Strain Curve

Figure 2 shows the stress–strain curve of sandstone under triaxial loading under Na_2_SO_4_ corrosion. Due to space limitations, only the stress–strain curves of *α* = 0.02 mol·L^−^^1^ at pH = 2 and 12 and the stress–strain curves of pH = 2 at *α* = 0.04 mol·L^−^^1^ and 0.10 mol·L^−^^1^ are given. Under the action of Na_2_SO_4_ corrosion, the changing trend of the triaxial loading stress–strain curve of sandstone is roughly the same as that of typical rock, which can be divided into four stages: (1) In the compaction stage, the curve is concave upward, the original micro-cracks in the rock are gradually compacted, the slope of the curve increases gradually, and the stiffness increases significantly. (2) In the elastic stage, the curve is straight and the specimen is further compressed, but the mechanical properties remain unchanged. (3) In the plastic yield stage, the curve is concave downward, the stress–strain curve gradually changes from linear to nonlinear, the new cracks continue to expand and develop, and the micro-cracks transition to macro cracks. (4) In the strain-softening stage, when the stress reaches the bearing limit, a macro fracture surface appears on the specimen, the bearing capacity decreases rapidly, and the deformation increases rapidly. When the confining pressure is low, the sandstone shows brittle failure. With the increase of confining pressure, the failure mode gradually transitions from brittleness to ductility, and the residual strength gradually increases.

In order to better observe the influence of Na_2_SO_4_ corrosion on sandstone, take the confining pressure of 10 MPa as an example and draw the axial stress–strain curve of sandstone under different pHs and concentrations, as shown in Figure 3. At the same concentration, with the gradual increase of pH of Na_2_SO_4_ solution, the compaction stage and plastic yield stage of the sandstone stress–strain curve are shortened first and then extended, the elastic stage is extended first and then shortened, and the slope of the curve in the strain-softening stage is increased first and then decreased. At the same pH, with the gradual increase of Na_2_SO_4_ solution concentration, the compaction stage and plastic yield stage of the sandstone stress–strain curve gradually extend, the elastic stage gradually shortens and the curve slope in the strain-softening stage gradually decreases. The damage and deterioration of sandstone under Na_2_SO_4_ corrosion are significant. The reason is that the Na_2_SO_4_ solution reacts with the active minerals on the surface of the sandstone, which destroys the integrity of the specimen surface and makes the particles loose and easy to peel off. The solution continues to penetrate along the initial micro-cracks of the specimen, resulting in the gradual increase of the specimen pores and the gradual penetration between the cracks, which finally reduces the overall mechanical properties. According to the triaxial loading test results, taking the confining pressure of 10 MPa as an example, the mechanical parameters of Na_2_SO_4_ corroded sandstone with different pHs and concentrations *α* are shown in Table 2.

### 3.2. Analysis of Mechanical Parameter

Figure 4 shows the distribution curve of mechanical parameters of sandstone with a pH of the Na_2_SO_4_ solution under different confining pressures. The peak strength, peak strain, and elastic modulus of sandstone first increase and then decrease with pH, and the maximum value is taken at pH = 7. The Poisson’s ratio first decreases and then increases with pH, and the minimum value is also taken at pH = 7. According to Table 2, when the pH = 7, the peak strength is 146.83 MPa, the peak strain is 0.8654%, the elastic modulus is 17.98 GPa, and the Poisson’s ratio is 0.293. When pH = 2, the peak strength is 115.71 MPa, the peak strain is 0.7351%, the elastic modulus is 16.07 GPa, and the Poisson’s ratio is 0.371. When the pH is reduced from 7 to 2, the peak strength reduced by 26.89%, the peak strain reduced by 17.73%, the elastic modulus reduced by 11.89%, and the Poisson’s ratio increased by 26.62%. The stronger the acidity of the Na_2_SO_4_ solution, the more serious the corrosion effect on sandstone.

Figure 5 shows the distribution curve of mechanical parameters of sandstone with concentration *α* of Na_2_SO_4_ solution under different confining pressures. The peak strength, peak strain, and elastic modulus of sandstone decrease gradually with the concentration, while Poisson’s ratio shows the opposite trend. According to Table 2, when *α* = 0.02 mol·L^−^^1^, the peak strength is 115.71 MPa, the peak strain is 0.7351%, the elastic modulus is 16.07 GPa, and the Poisson’s ratio is 0.371. When *α* = 0.10 mol·L^−^^1^, the peak strength is 86.54 MPa, the peak strain is 0.6197%, the elastic modulus is 14.37 GPa, and the Poisson’s ratio is 0.415. The higher the concentration of the Na_2_SO_4_ solution, the more serious the corrosion of sandstone.

According to the above parameter analysis, the pH and concentration *α* of the Na_2_SO_4_ solution both have an impact on the mechanical properties of sandstone. At the same concentration, sandstone is damaged to varying degrees in an acidic and alkaline environment, and the stronger the acidity or alkalinity is, the more serious the damage is. By observing the curve distribution law in Figure 4, the relationship between each parameter and pH approximately meets the quadratic function. According to the parameter calculation results, the influence degree of different pHs on the mechanical parameters is as follows: strong acid environment (pH ≤ 4) > strong alkali environment (pH ≥ 10) > weak acid environment (4 ≤ pH < 6) > weak alkali environment (8 ≤ pH < 10) > neutral environment (6 ≤ pH < 8). At the same pH, the corrosion degree of sandstone increases gradually with the increase of concentration. By observing the curve distribution in Figure 5, the relationship between each parameter and concentration is an approximately exponential function, and the greater the concentration, the greater the slope of the curve.

According to Mohr–Coulomb strength criteria, the relationship between principal stresses can be expressed as:(1)$$\sigma_{1}=A\sigma_{3}+B$$
where *σ*_1_ is axial stress, *σ*_3_ is confining pressure, *A* and *B* are material parameters, where *A* = (1 + sin *φ*)/(1 − sin *φ*), *B* = 2*c*cos *φ*/(1 − sin *φ*), *c* is cohesion, and *φ* is internal friction angle. The triaxial loading test results of Na_2_SO_4_ corroded sandstone are fitted through Equation (1), the material parameters *A* and *B* are obtained, and then the cohesion and internal friction angle are obtained according to Equation (2):(2)$$\left\{\begin{array}{l} \varphi =\arcsin\frac{A-1}{A+1}\\ c=\frac{B}{2}\frac{1-\sin\varphi }{\cos\varphi } \end{array}\right.$$

Figure 6 shows the distribution curve of cohesion *c* and internal friction angle *φ* with pH and concentration *α* of Na_2_SO_4_ solution. At the same concentration, with the gradual increase of pH, the cohesion and internal friction angle increase first and then decrease, and the maximum value is taken at pH = 7. The relationship between shear strength parameters and pH approximately satisfies the quadratic function. At the same pH, with the gradual increase of concentration, the cohesion and internal friction angle of sandstone show a gradually decreasing trend, and the greater the concentration, the greater the slope of the curve. According to Figure 6, when pH = 7, the cohesion and internal friction angle are 17.26 MPa and 42.23° respectively. When pH = 2, the cohesion is 14.41 MPa, the internal friction angle is 39.02°, the pH is reduced from 7 to 2, and the two parameters are reduced by 16.51% and 7.60% respectively. When pH = 12, the cohesion is 14.95 MPa, the internal friction angle is 39.95°, the pH increases from 7 to 12, and the decreases of two parameters are 13.38% and 5.39% respectively. When sandstone is in a strong acid environment, the corrosion degree is the most serious, followed by a strong alkaline environment. When sandstone is in weak acid or weak alkaline environment, the corrosion degree is alleviated, and when sandstone is in a neutral environment, the corrosion degree is the least.

### 3.3. Analysis of Characteristic Stress

The loading failure of rock has obvious stage characteristics. The characteristic stress can be used as the basis for the division of rock loading failure stage, including the closing stress *σ*_cc_ characterizing the compaction of micro cracks, the crack initiation stress *σ*_ci_ formed by micro-cracks, the dilatancy stress *σ*_cd_ of rock entering the plastic yield stage due to volume expansion, and the peak strength *σ*_pk_. Among them, initiation stress *σ*_ci_ and dilatancy stress *σ*_cd_ are two important indexes to describe the development and expansion of cracks in rocks. The crack volume strain curve can reflect the change law of crack closure and opening during rock loading, so the crack volume strain method is used to determine the characteristic stress in this paper.

In the triaxial loading test, the volume strain can not be measured directly. It needs to be calculated through the relationship between volume strain and axial and radial strain. The three meet the following relationship:(3)$$\varepsilon_{v}=\varepsilon_{1}+2\varepsilon_{3}$$
where *ε*_v_ is volume strain, *ε*_1_ and *ε*_3_ are axial strain and radial strain respectively.

Volume strain can also be expressed by the following equation:(4)$$\varepsilon_{v}=\varepsilon_{v}^{e}+\varepsilon_{v}^{c}$$
where *ε*_v_^e^ is elastic volume strain, *ε*_v_^c^ is the crack volume strain.

According to Hook’s law, the elastic volume strain under conventional triaxial loading can be expressed as:(5)$$\varepsilon_{v}^{e}=\varepsilon_{1}^{e}+2\varepsilon_{3}^{e}=\frac{1-2v}{E}\left(\sigma_{1}+2\sigma_{3}\right)$$

According to Equations (3)–(5), the crack volume strain can be expressed as:(6)$$\varepsilon_{v}^{c}=\varepsilon_{v}-\varepsilon_{v}^{e}=\varepsilon_{1}+2\varepsilon_{3}-\frac{1-2v}{E}\left(\sigma_{1}+2\sigma_{3}\right)$$
where *E* is the elastic modulus, *v* is Poisson’s ratio.

All parameters in Equation (6) can be obtained from the triaxial loading test. According to the above equations, the crack volume strain of sandstone under different test conditions can be obtained. Figure 7 shows the stress–strain curve, volume–strain curve, and crack volume–strain curve of typical rock during the whole process of triaxial loading, and *σ*_cr_ is the residual strength [27].

According to the above method, the characteristic stress of sandstone under Na_2_SO_4_ corrosion is obtained, as shown in Table 3. At the same concentration, the closure stress *σ*_cc_, initiation stress *σ*_ci_, and dilatancy stress *σ*_cd_ first increase and then decrease with pH, and the maximum value is taken at pH = 7. When pH = 7, the closure stress *σ*_cc_ is 44.58 MPa, the initiation stress *σ*_ci_ is 74.69 MPa, and the expansion stress *σ*_cd_ is 107.19 MPa, the ratios of *σ*_cc_/*σ*_pk_, *σ*_ci_/*σ*_pk_, and *σ*_cd_/*σ*_pk_ are 0.3115, 0.5020, and 0.7043, respectively, and *σ*_ci_/*σ*_cd_ is 0.3115. When pH = 2, the three characteristic stresses are 36.04, 58.09, and 81.50 MPa respectively, the ratios of *σ*_cc_/*σ*_pk_, *σ*_ci_/*σ*_pk_, and *σ*_cd_/*σ*_pk_ are 0.3115, 0.5020, and 0.7043, respectively, and *σ*_ci_/*σ*_cd_ is 0.3115. The pH goes from 7 to 2, the three characteristic stresses decreased by 23–32%, and the characteristic stress ratio had no obvious change law. When pH = 12, the three characteristic stresses are 37.89, 62.65, and 90.48 MPa, respectively, the ratios of *σ*_cc_/*σ*_pk_, *σ*_ci_/*σ*_pk_, and *σ*_cd_/*σ*_pk_ are 0.3042, 0.5029, and 0.7263 respectively, and *σ*_ci_/*σ*_cd_ is 0.3042. The pH rises from 7 to 12, the three characteristic stresses decrease by 17–20%, and the characteristic stress ratio also has no obvious change law.

At the same pH, the closure stress *σ*_cc_, initiation stress *σ*_ci_, and dilatancy stress *σ*_cd_ of sandstone decrease gradually with the concentration. When *α* = 0.02 mol·L^−^^1^, the three characteristic stresses are 36.04, 58.09, and 81.50 MPa respectively, the ratios of *σ*_cc_/*σ*_pk_, *σ*_ci_/*σ*_pk_, and *σ*_cd_/*σ*_pk_ are 0.3115, 0.5020 and 0.7043, respectively, and *σ*_ci_/*σ*_cd_ is 0.3115. When *α* = 0.10 mol·L^−^^1^, the three characteristic stresses are 30.84, 44.31, and 67.72 MPa, respectively, the ratios of *σ*_cc_/*σ*_pk_, *σ*_ci_/*σ*_pk_, and *σ*_cd_/*σ*_pk_ are 0.3564, 0.5120 and 0.7825 respectively, and *σ*_ci_/*σ*_cd_ is 0.3564. The concentration increased from 0.02 mol·L^−^^1^ to 0.1 mol·L^−^^1^, the three characteristic stresses decreased by 8–20%, but the characteristic stress ratio had no obvious change law.

According to the above analysis, the characteristic stress of sandstone is seriously affected by the Na_2_SO_4_ solution, while the characteristic stress ratio has no obvious change law. The reason can be explained that the characteristic stress is the stress level of rock under a specific stress state, and it is the mechanical parameter of the rock itself, and it is affected by the external environment and loading state. *σ*_cc_/*σ*_pk_, *σ*_ci_/*σ*_pk_, and *σ*_ci_/*σ*_cd_ are the ratios between characteristic stresses, which is the mechanical embodiment of the microstructure properties of rock. For example, *σ*_cc_/*σ*_pk_ can characterize the initial porosity in the rock, *σ*_ci_/*σ*_pk_ can characterize the anisotropy of rock *σ*_cd_/*σ*_pk_ can characterize the anti-deformation ability of rock. The three are little affected by the external environment, and the value of the same kind of rock usually fluctuates in a certain range.

According to the triaxial loading test results of Na_2_SO_4_ corroded sandstone, the distribution curve of characteristic stress with pH and concentration *α* of Na_2_SO_4_ solution under different confining pressures are shown in Figure 8. By observing the change law of Figure 8a–c, it is found that the closing stress *σ*_cc_, initiation stress *σ*_ci_, and dilatancy stress *σ*_cd_ approximately meet the quadratic function relationship with pH, and the corrosive effect of acidic environment on sandstone is stronger than that of an alkaline environment. By observing the change of Figure 8d–f, it is found that the closing stress *σ*_cc_, initiation stress *σ*_ci_, and dilatancy stress *σ*_cd_ of sandstone approximately meet the exponential function relationship with the concentration *α*, and the greater the concentration, the greater the slope of the curve, and the stronger the corrosion effect on sandstone.

### 3.4. Analysis of Energy Mechanism

The deformation and failure process of rock is essentially the mutual transformation of energy. Energy accumulation and dissipation are the real causes of rock failure. The energy method is to study the compression failure process of rock in essence. This method can reflect the characteristics of rock macro failure, strength weakening, micro-crack development, and evolution. In underground engineering, the energy analysis of the rock failure process has attracted much attention. Analyzing the energy transformation law in the process of rock loading failure and revealing the mechanical properties and failure mechanism of rock from the perspective of energy is of great significance to understanding the essential properties of rock [28].

Assuming that the heat energy generated by the temperature change of the external system is not considered, the work done by the external force on the rock can be roughly divided into two parts, one part is stored in the form of elastic strain energy, and the other part is dissipated in the form of damage energy. Considering the deformation of a single specimen under external force, assuming that there is no heat exchange with the outside, the system is considered to be a closed system [29]. The energy satisfies the following equation:(7)$$U=U^{d}+U^{e}$$
where *U* is the total energy generated by work done by an external force, *U*^d^ is the dissipated energy, and *U*^e^ is the elastic strain energy.

The calculation principle of rock energy is shown in Figure 9. In the initial stage (OA), almost all the external energy is used for micro crack compaction. When the stress reaches A, the micro-cracks in the rock are basically closed, and the external energy begins to be used for the elastic deformation between particle frameworks (AB). With the continuous input of energy, it is used for elastic deformation where particle frameworks gradually increase. When the rock reaches the elastic deformation limit, the external energy begins to be used for crack initiation, expansion, and particle slip (BC). When the stress reaches the bearing limit, the rock produces macro cracks, and the energy stored before the peak is released instantaneously, and the rock is damaged (CD). In Figure 9, the area OABCG is the total work done by an external force, marked as *U*. The area OABCF is the dissipated energy before rock failure, marked as *U*^d^. The area CFG is the elastic strain energy stored before rock failure, marked as *U*^e^. CF is the unloading elastic modulus *E*_i_, which can be replaced by the loading elastic modulus *E*_0_.

According to Equation (7), the dissipated energy can be expressed as:(8)$$U^{d}=U-U^{e}$$

Under conventional triaxial loading (*σ*_1_ > *σ*_2_ = *σ*_3_), the elastic strain energy *U*^e^ can be expressed as:(9)$$U^{e}=\frac{1}{2E_{0}}\left[\sigma_{1}^{2}+2\sigma_{3}^{2}-2\mu_{0}\left(2\sigma_{1}\sigma_{3}+\sigma_{3}^{2}\right)\right]$$

The dissipated energy *U*^d^ can be expressed as:(10)$$U^{d}=\int_{0}^{\varepsilon_{1}}\sigma_{1}d\varepsilon_{1}-\frac{1}{2E_{0}}\left[\sigma_{1}^{2}+2\sigma_{3}^{2}-2\mu_{0}\left(2\sigma_{1}\sigma_{3}+\sigma_{3}^{2}\right)\right]$$

According to Equations (7)–(10), combined with the triaxial loading test data, the energy evolution curves of sandstone under different test conditions can be obtained, as shown in Figure 10. The total strain energy *U* and dissipated energy *U*^d^ increase gradually with the axial strain, while the elastic strain energy *U*^e^ increases first and then decreases with the axial strain. Take pH = 12, *α* = 0.02 mol·L^−^^1^, for example

(1)Before the closure stress *σ*_cc_, the three kinds of energy increase slowly, because the initial crack is compressed and closed, the initial stiffness is relatively small and the energy conversion rate is low.(2)When the axial stress exceeds the closure stress *σ*_cc_, the specimen comes to the elastic deformation stage, the total energy and elastic strain energy gradually increase, and the two values are close, and the dissipated energy remains basically unchanged and always remains at a low value. The reason is that before the expansion stress *σ*_cd_, the damage degree is light, and the work done by the external force is mainly transformed into the elastic strain energy stored in the specimen.(3)When the axial stress exceeds the dilatancy stress *σ*_cd_, the specimen enters the unstable crack propagation stage. The damage degree and plastic deformation increase greatly, the dissipated energy begins to accelerate, the curve slope gradually increases, and the slope of the elastic strain energy curve begins to decrease, but the elastic strain energy is still the main stage. When the axial stress reaches the peak strength *σ*_pk_, the elastic strain energy releases rapidly, the dissipated energy increases rapidly.

Taking the confining pressure of 10 MPa as an example, the total energy, elastic strain energy, and dissipated energy corresponding to the peak strength of Na_2_SO_4_ corroded sandstone are shown in Table 4. At the same concentration, with the gradual increase of pH, the total energy and elastic strain energy corresponding to the peak strength first increase and then decrease, while the dissipated energy is opposite to the former. When pH = 7, the three energies corresponding to the peak strength are 80.51 kJ·m^−^^3^, 74.06 kJ·m^−^^3^, and 6.45 kJ·m^−^^3^ respectively. When pH = 2, the three energies corresponding to the peak strength are 51.94 kJ·m^−^^3^, 43.63 kJ·m^−^^3^, and 8.31 kJ·m^−^^3^ respectively. When pH decreases from 7 to 2, the total energy and elastic strain energy decrease by 55.01% and 69.75% respectively, while the dissipated energy increases by 28.84%. When pH = 12, the three energies corresponding to the peak strength are 62.95 kJ·m^−^^3^, 53.26 kJ·m^−^^3^, and 8.69 kJ·m^−^^3^ respectively. When pH rises from 7 to 12, the total energy and elastic strain energy decrease by 27.89% and 39.05% respectively, while the dissipated energy increases by 34.72%. At the same pH, with the gradual increase of concentration, the total energy, elastic strain energy, and dissipated energy corresponding to the peak strength show a decreasing trend. When concentration *α* = 0.02 mol·L^−^^1^, the three energies are 51.94 kJ·m^−^^3^, 43.63 kJ·m^−^^3^, and 8.31 kJ·m^−^^3^ respectively. When concentration *α* = 0.10 mol·L^−^^1^, the three energies are 30.10 kJ·m^−^^3^, 23.78 kJ·m^−^^3^, and 6.32 kJ·m^−^^3^ respectively. The concentration increased from 0.02 mol·L^−^^1^ to 0.1 mol·L^−^^1^, and the three energies decreased by 72.56, 83.47, and 31.49% respectively.

Through the above analysis, the strong acid and strong alkali environment have a significant impact on the energy damage of sandstone. The corrosion effect of acid environment on sandstone is significantly stronger than that of alkali and the greater the concentration, the stronger the corrosion to sandstone. The reason is that the corrosion of Na_2_SO_4_ reduces the mechanical properties of sandstone, reduces the bearing capacity, decreases the conversion between work and energy, and increases the energy consumed by friction and sliding between sandstone skeleton and particles, resulting in the decrease of total energy and elastic strain energy and the increase of dissipated energy. According to Table 4, although the proportion of elastic energy and the proportion of dissipated energy change regularly with the pH and concentration of Na_2_SO_4_ solution, the increase is small. The elastic strain energy accounts for about 85.5% of the total energy, and the dissipated energy accounts for about 14.5% of the total energy. The energy evolution law of Na_2_SO_4_ corroded sandstone under different confining pressures is basically the same, which will not be repeated.

Rock failure is a comprehensive process of energy transformation, and energy dissipation *U*^d^ reflects the process of crack initiation, expansion, and final failure. Therefore, the ratio of the dissipated energy at any time to the dissipated energy *U*^d^_peak_ corresponding to the peak strength is defined as the damage variable *D*, and the expression is as follows:(11)$$D=\frac{U^{d}}{U_{\mathrm{peak}}^{d}}$$
where *D* is the damage variable, *U*^d^ is the dissipated energy at any time, and *U*^d^_peak_ is the dissipated energy corresponding to the peak strength.

According to Equation (11) and the characteristics of the dissipated energy evolution curve, when the dissipated energy *U*^d^ exceeds the dissipated energy *U*^d^_peak_ corresponding to the peak strength at any time, the damage variable *D* > 1, and the damage variable loses its significance. Therefore, this paper only analyzes the damage of sandstone before peak strength. Taking the confining pressure of 10 MPa as an example, the distribution curve of damage variables under different pHs and concentrations *α* is shown in Figure 11. With the increase of axial strain, the damage variable increases under different test conditions. According to Figure 11a, when pH = 7, the damage variable corresponding to the same strain is the smallest. According to Figure 11b, when *α* = 0.02 mol·L^−^^1^, the damage variable corresponding to the same strain is the smallest. It shows that a neutral environment and low concentration have little damage to sandstone before failure.

## 4. Establishment and Verification of Damage Equation

In order to more accurately describe the corrosion effect of different concentrations of Na_2_SO_4_ solution on sandstone, the damage evolution equation considering different concentrations is established in this paper.

The damage state of rock can be expressed by the following equation:(12)$$Y=\frac{U^{e}}{1-D}$$
where *Y* is the damage energy dissipation rate, *U*^e^ is the elastic strain energy, and *D* is the damage variable, which can be obtained by Equation (11).

The rock damage variable can also be expressed by Equation (13) [30]:(13)$$D=1-\exp\left[-B{\left|Y-Y_{0}\right|}^{\frac{1}{n}}\right]$$
where *B*, *n*, and *Y*_0_ are rock material parameters, depending on rock properties.

Take two logarithms on both sides of Equation (13):(14)$$\ln\left[-\ln\left(1-D\right)\right]=\ln B+\frac{1}{n}\ln\left|Y-Y_{0}\right|$$

Assuming that the specimen has no initial damage before loading (i.e., *Y*_0_ = 0), Equation (14) can be rewritten as:(15)$$\ln\left[-\ln\left(1-D\right)\right]=\ln B+\frac{1}{n}\ln Y$$

Let *x* = ln*Y*, *y* = ln[−ln(1−*D*)], then Equation (15) can be expressed as:(16)$$y=\frac{1}{n}x+\ln B$$

Obviously, Equation (16) is a linear function of *y* with respect to *x*. Let *λ* = 1/*n*, *β* = ln*B*, Equation (16) can be expressed as:(17)y=λx+β

According to the triaxial loading test results of corroded sandstone with different concentrations of Na_2_SO_4_, the damage energy dissipation rate *Y* and damage variable *D* can be calculated, and then a group of (*x*, *y*) data can be obtained. The fitting parameter *λ* and *β* can be obtained by linear fitting with Equation (17), and then the parameters *B* and *n* can be obtained from Equation (18):(18)$$\left\{\begin{array}{l} n=\frac{1}{\lambda }\\ B=\exp\left(\beta \right) \end{array}\right.$$

Figure 12 shows (*x*, *y*) data and fitting curves under different concentrations when the confining pressure is 10 MPa and pH = 2. The test curve is close to the fitting curve, and the correlation coefficients are more than 0.9, indicating that Equation (13) basically conforms to the test results. The calculation results of parameters *λ* and *β* and parameters *B* and *n* in Equation (13) under different concentrations are shown in Table 5.

The distribution curve of parameters *B* and *n* with the concentration *α* of Na_2_SO_4_ are shown in Figure 13.

The parameters *B* and *n* decreased gradually with the concentration *α*. By fitting the curve with Origin, it is found that the relationship between parameters *B*, *n*, and concentration *α* meets the negative exponential function, which can be expressed by the following equation:(19)$$\left\{\begin{array}{l} B=M_{1}+N_{1}\exp(-k_{1}\alpha )\\ n=M_{2}+N_{2}\exp(-k_{2}\alpha ) \end{array}\right.$$
where *M*_1_, *M*_2_, *N*_1_, *N*_2_, *k*_1_, and *k*_2_ are fitting parameters. By substituting Equation (19) into Equation (13), the energy damage evolution equation of sandstone considering the effect of Na_2_SO_4_ concentration *α* can be obtained:(20)$$D=1-\exp\left[-[M_{1}+N_{1}\exp(-k_{1}\alpha )]{\left|Y-Y_{0}\right|}^{\frac{1}{\left[M_{2}+N_{2}\exp(-k_{2}\alpha )\right]}}\right]$$

In order to verify the rationality of the energy damage evolution equation considering the Na_2_SO_4_ concentration effect, the fitting parameters are substituted into Equation (20). The comparison of the test curve, the theoretical curve in this paper, and the theoretical curve in reference [30] is shown in Figure 14. Both the theoretical curve in this paper and the theoretical curve in reference [30] can describe the damage evolution law of sandstone, but the theoretical curve in this paper is closer to the test curve, which shows that the energy damage evolution equation considering Na_2_SO_4_ concentration effect can better describe the damage characteristics of sandstone loading failure process, and verify the rationality and accuracy of the model in this paper.

## 5. Analysis of Corrosion Mechanism of Na_2_SO_4_

Under the action of Na_2_SO_4_ corrosion, SO_4_^2−^ ions in the solution react with the active mineral components on the surface of sandstone, resulting in the dissolution and migration of particles on the rock surface, resulting in the increase of porosity and heterogeneity of sandstone. Through X-ray, the main chemical components of sandstone in this paper include quartz (SiO_2_), calcite (CaCO_3_), and hematite (Fe_2_O_3_), as well as traces of dolomite, chlorite, and other substances. Among them, quartz (SiO_2_) reacts with Na_2_SO_4_ to produce water-insoluble mirabilite (Na_2_SiO_3_). The chemical equation is as follows:SiO_2_ + Na_2_SO_4_ = Na_2_SiO_3_ + SO_3_

Ca^2+^ in calcite and Fe^2+^ in hematite can exist in free form in Na_2_SO_4_ solution. According to the ion concentration test results, the distribution curves of Ca^2+^ and Fe^2+^ concentrations with damage variables under different test conditions are shown in Figure 15. With the increasing concentration of Ca^2+^ and Fe^2+^ in the solution, the damage variable increases gradually, and the deterioration degree of sandstone increases gradually. The relationship between the concentration of two ions and the damage variable approximately satisfies a linear function.

## 6. Discussion

As a kind of natural geological body, the physical and mechanical properties of rock mainly depend on the mineral composition and the connection between particles. There are both physical and chemical interactions between chemical solutions and rocks. In physics, the dissolution of water on rock leads to the decrease of interconnecting force between mineral particles and friction force, and the pore pressure of water reduces the effectiveness of confining pressure, thus producing a splitting effect on micropores. In chemistry, chemical corrosion changes the microscopic structure and mineral composition inside the rock weakens the connection between mineral particles or crystals and makes the porosity increase and become soft, leading to changes in its mechanical properties. The combined effects of these two aspects eventually lead to changes in the physical and mechanical properties of rock. In this paper, a Na_2_SO_4_ solution is used to simulate the chemical environment of sandstone, in which the products generated by the interaction between SO_4_^2−^ and sandstone accumulate in the pores. These expansive products absorb water and increase the volume, resulting in the increase of pore water pressure, destroying the internal structure and further infiltrating the chemical solution, which leads to the deterioration of mechanical properties of the specimen, and the stronger the solubility and acidity (or alkalinity) of the solution, the more serious the damage and deterioration of sandstone.

Rock weathering is a very complex geological process, and the influencing factors are complex and diverse, among which chemical corrosion is one of the important factors leading to rock weathering and destruction. In a dry environment, physical weathering is dominant, but in a chemical solution environment, chemical corrosion weathering is dominant. Sandstone is usually subjected to weathering. When sandstone is subjected to chemical corrosion and dry weathering alternately, its deterioration rate will be further accelerated, which seriously affects the safety and stability of the surrounding rock of underground engineering.

In addition, underground projects usually have to be in service for a long time, ranging from years to decades. Creep and cracking occur in rocks under long-term load. After chemical corrosion, the mechanical properties of rock decrease significantly. Under the same load, the long-term strength of rock decreases significantly, creep deformation increases, durability decreases, and service time is greatly shortened. At the same time, after chemical corrosion, the cementation ability between particles decreases. Under the same load, the rock is more prone to cracking, which reduces the bearing capacity, and ultimately affects the long-term stability of underground engineering. The interaction between chemical corrosion and rock weathering, creep, and cracking should be fully considered during construction to reduce construction risk and cost.

## 7. Conclusions

Based on the triaxial loading test of sandstone corroded by Na_2_SO_4_, the basic mechanical parameters, characteristic stress, and energy evolution law under different pHs and concentrations were analyzed, the damage evolution equation considering the concentration of Na_2_SO_4_ solution was established, and the corrosion mechanism of sandstone in a Na_2_SO_4_ solution was analyzed. The main conclusions are as follows:With the gradual increase of pH in the Na_2_SO_4_ solution, the basic mechanical parameters (peak strength, peak strain, elastic modulus, cohesion, and internal friction angle) and characteristic stress parameters (closure stress, initiation stress, and dilatancy stress) of sandstone show the variation law of quadratic function, and the extreme values are taken when pH = 7. The influence degree of different pHs on the mechanical parameters is as follows: strong acid environment (pH ≤ 4) > strong alkali environment (pH ≥ 10) > weak acid environment (4 ≤ pH < 6) > weak alkali environment (8 ≤ pH < 10) > neutral environment (6 ≤ pH < 8). With the increasing concentration of the Na_2_SO_4_ solution, the basic mechanical parameters and characteristic stress parameters of sandstone change monotonically.With the gradual increase of pH in the Na_2_SO_4_ solution, the total energy and elastic strain energy of sandstone first increase and then decrease, while the dissipated energy shows the opposite trend. With the increasing concentration of the Na_2_SO_4_ solution, the three energies decreased gradually. Under different test conditions, the proportion of elastic energy and dissipated energy also change regularly with the pH and concentration of the Na_2_SO_4_ solution, but the change range is small.Under the same strain value, the damage variable of sandstone decreases first and then increases with the pH of the Na_2_SO_4_ solution, and gradually increases with the concentration. Based on the energy theory, the damage evolution equation considering the concentration of the Na_2_SO_4_ solution is established, and the rationality of the model is verified according to the test data.The Na_2_SO_4_ solution reacts with quartz (SiO_2_) on the surface of sandstone to form water-insoluble mirabilite (Na_2_SiO_3_). At the same time, Ca^2+^ in calcite and Fe^2+^ in hematite are dissolved and precipitated. With the increasing concentration of Ca^2+^ and Fe^2+^ in the solution, the damage variable increases gradually. The relationship between the two ion concentrations and the damage variable approximately satisfies a linear function.

## Figures and Tables

**Figure 1 materials-15-01613-f001:**
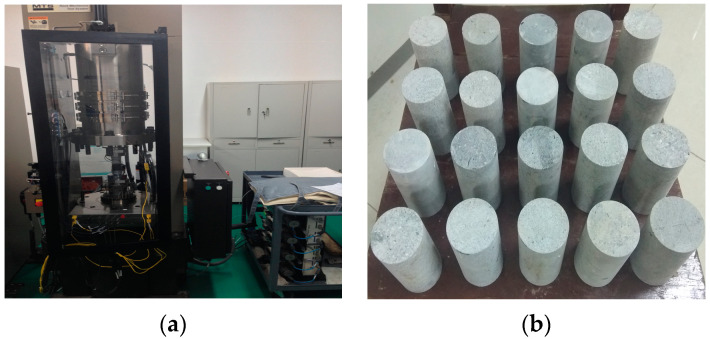
Test equipment and rock specimens. (**a**) MTS815.02 triaxial test system; (**b**) sandstone.

**Figure 2 materials-15-01613-f002:**
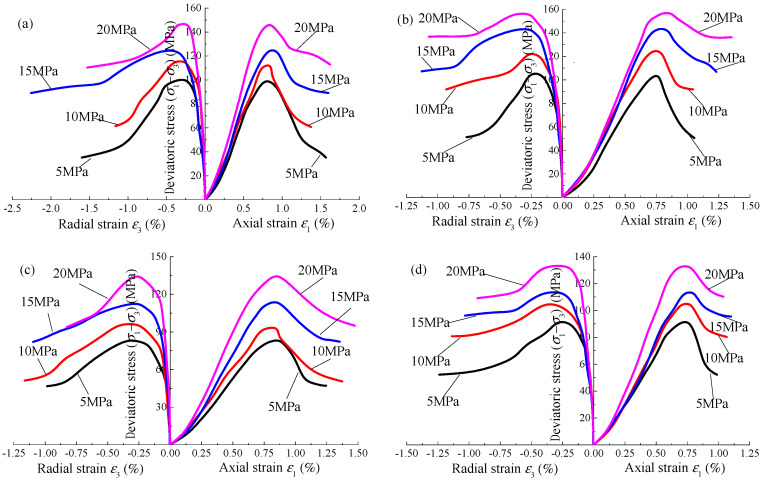
Stress–strain curves of Na_2_SO_4_ corroded sandstone under triaxial loading: (**a**) *α* = 0.02 mol·L^−1^, pH = 2; (**b**) *α* = 0.02 mol·L^−1^, pH = 12; (**c**) *α* = 0.04 mol·L^−1^, pH = 2; (**d**) 0.10 mol·L^−1^, pH = 2.

**Figure 3 materials-15-01613-f003:**
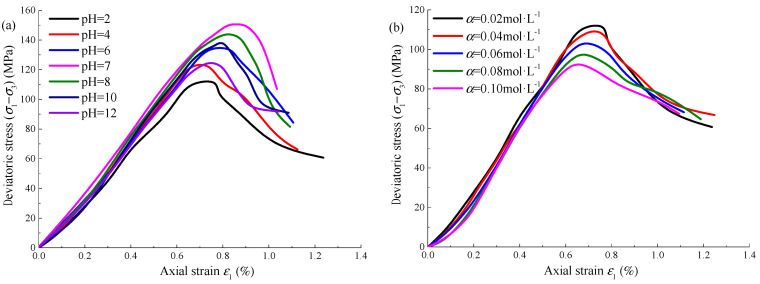
Axial stress–strain curve of Na_2_SO_4_ corroded sandstone: (**a**) *α* = 0.02 mol·L^−1^; (**b**) pH = 2.

**Figure 4 materials-15-01613-f004:**
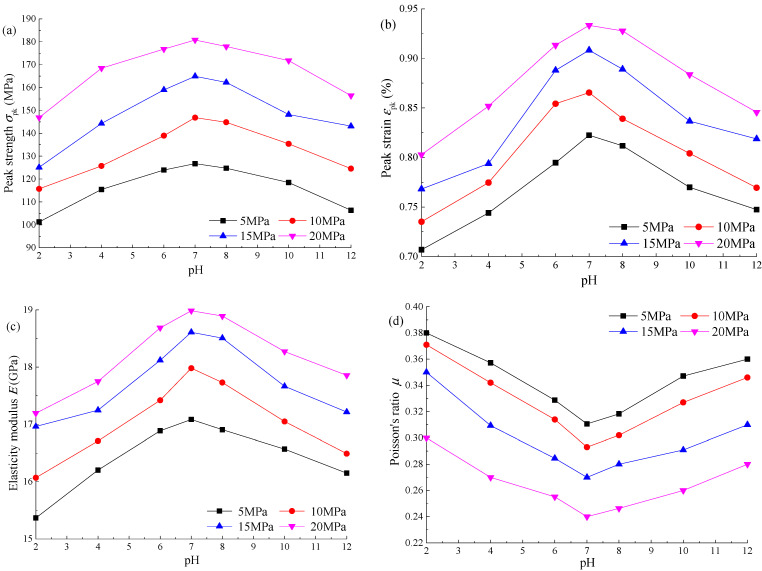
Distribution curve of mechanical parameters with pH of Na_2_SO_4_: (**a**) peak strength *σ*_pk_; (**b**) peak strain *ε*_pk_; (**c**) elasticity modulus *E*; (**d**) Poisson’s ratio *µ*.

**Figure 5 materials-15-01613-f005:**
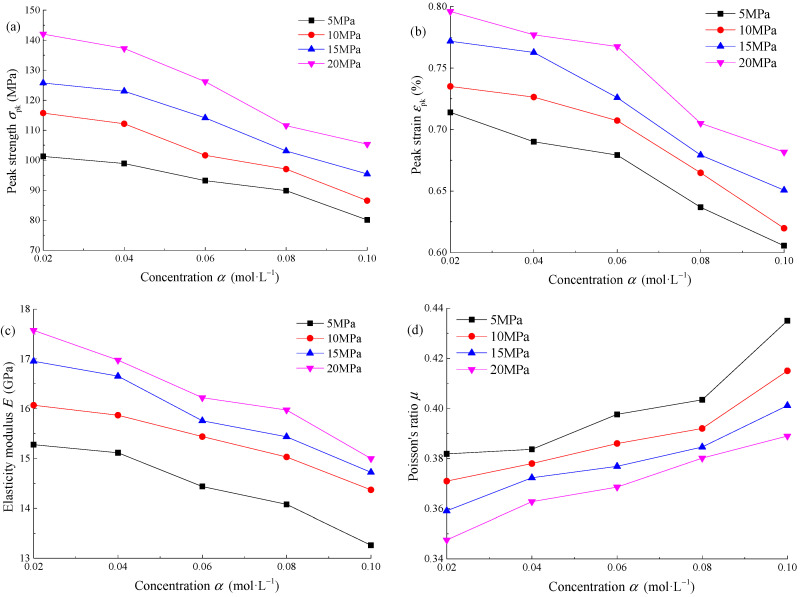
Distribution curve of mechanical parameters with concentration *α* of Na_2_SO_4_: (**a**) peak strength *σ*_pk_; (**b**) peak strain *ε*_pk_; (**c**) elasticity modulus *E*; (**d**) Poisson’s ratio *µ*.

**Figure 6 materials-15-01613-f006:**
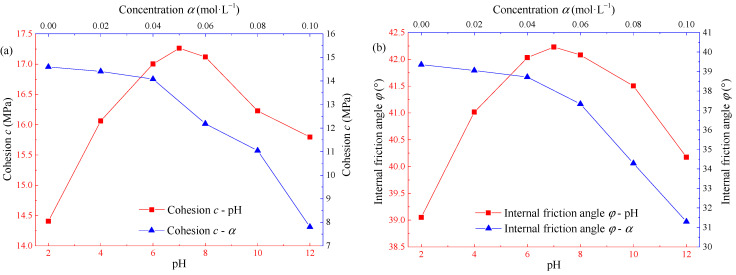
Distribution curve of shear strength parameters of Na_2_SO_4_ corroded sandstone: (**a**) cohesion *c*; (**b**) internal friction angle *φ*.

**Figure 7 materials-15-01613-f007:**
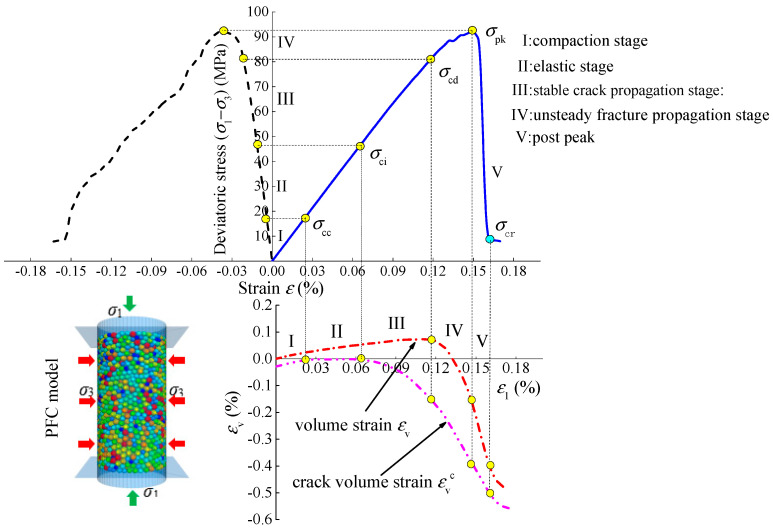
Curves of stress–strain, volume–strain, and crack volume–strain in the whole process of rock triaxial compression.

**Figure 8 materials-15-01613-f008:**
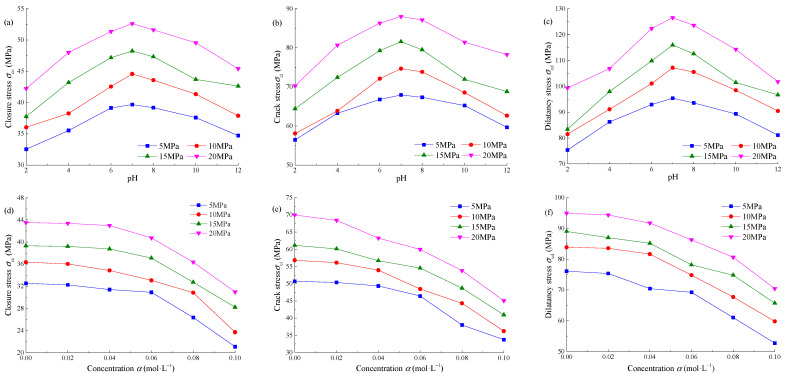
Characteristic stress distribution curve of Na_2_SO_4_ corroded sandstone: (**a**) closure stress *σ*_cc_-pH; (**b**) initiation stress *σ*_ci_−pH; (**c**) dilatancy stress *σ*_cd_−pH; (**d**) closure stress *σ*_cc_−*α*; (**e**) initiation stress *σ*_ci_−*α*; (**f**) dilatancy stress *σ*_cd_−*α*.

**Figure 9 materials-15-01613-f009:**
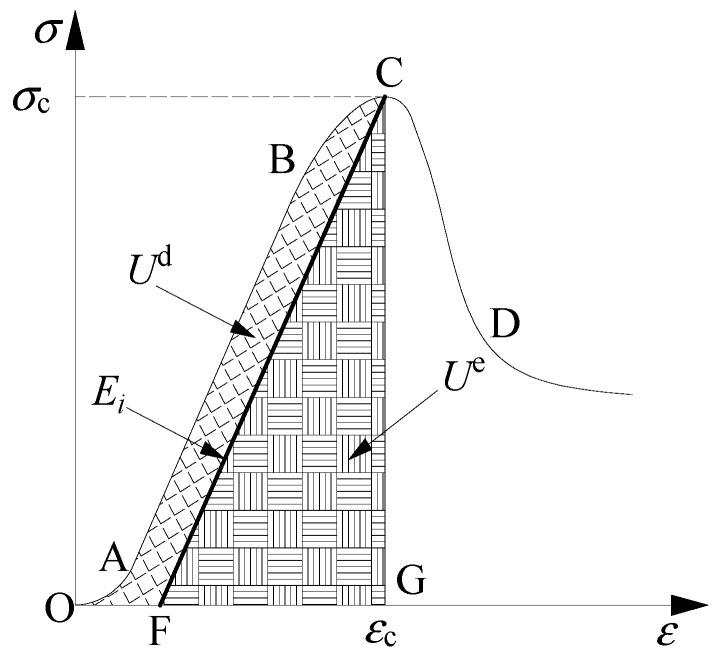
Principle of rock energy calculation.

**Figure 10 materials-15-01613-f010:**
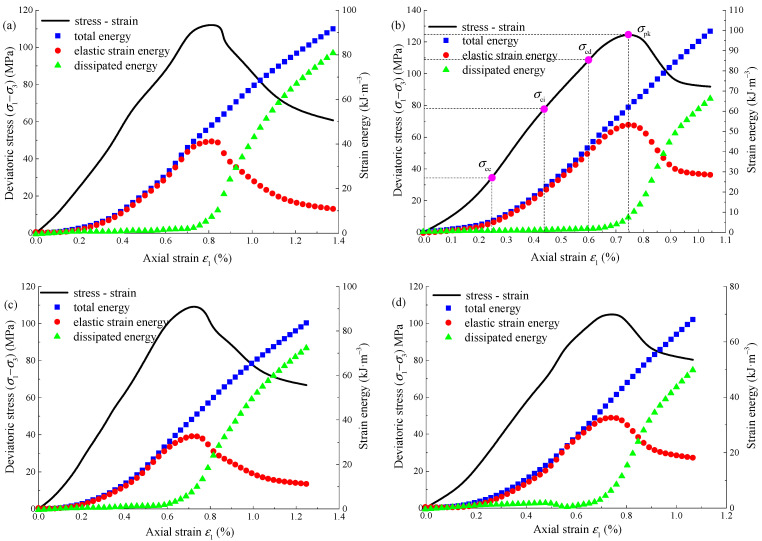
Energy evolution curve of Na_2_SO_4_ corroded sandstone: (**a**) *α* = 0.02 mol·L^−1^, pH = 2; (**b**) *α* = 0.02 mol·L^−1^, pH = 12; (**c**) *α* = 0.04 mol·L^−1^, pH = 2; (**d**) 0.10 mol·L^−1^, pH = 2.

**Figure 11 materials-15-01613-f011:**
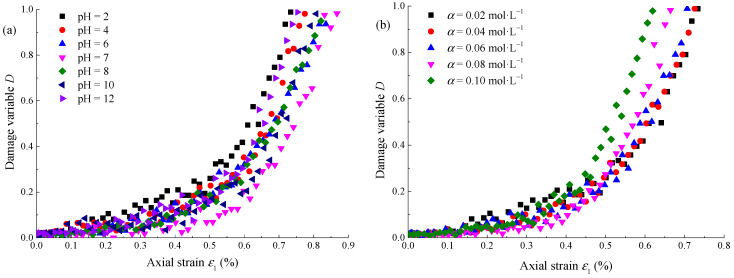
Damage variation evolution curve of Na_2_SO_4_ corroded sandstone: (**a**) *α* = 0.02 mol·L^−1^; (**b**) pH = 2.

**Figure 12 materials-15-01613-f012:**
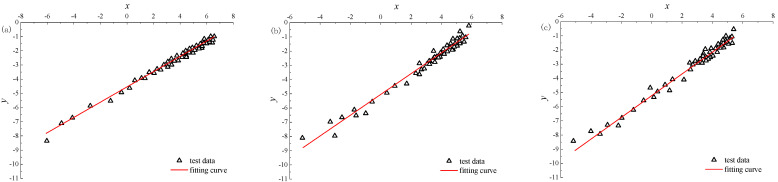
Data distribution curve of (*x*, *y*) of Na_2_SO_4_ at different concentrations: (**a**) *α* = 0.02 mol·L^−1^; (**b**) *α* = 0.06 mol·L^−1^; (**c**) *α* = 0.10 mol·L^−1^.

**Figure 13 materials-15-01613-f013:**
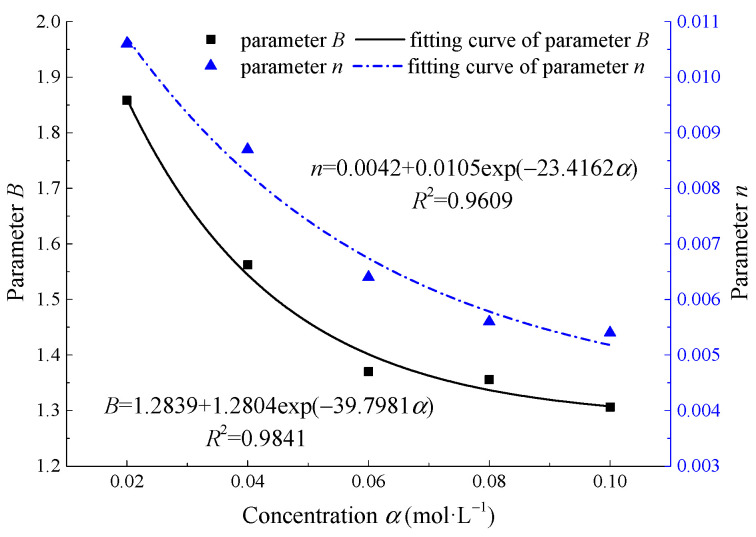
Relationship between parameters *B*, *n*, and Na_2_SO_4_ concentration *α*.

**Figure 14 materials-15-01613-f014:**
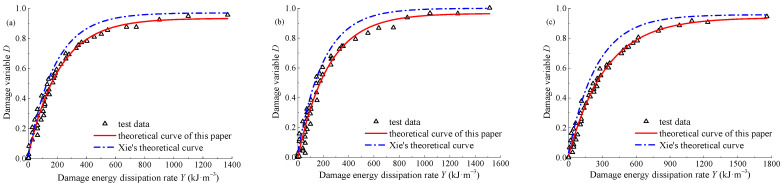
Model validation: (**a**) *α* = 0.02 mol·L^−1^; (**b**) *α* = 0.06 mol·L^−1^; (**c**) *α* = 0.10 mol·L^−1^.

**Figure 15 materials-15-01613-f015:**
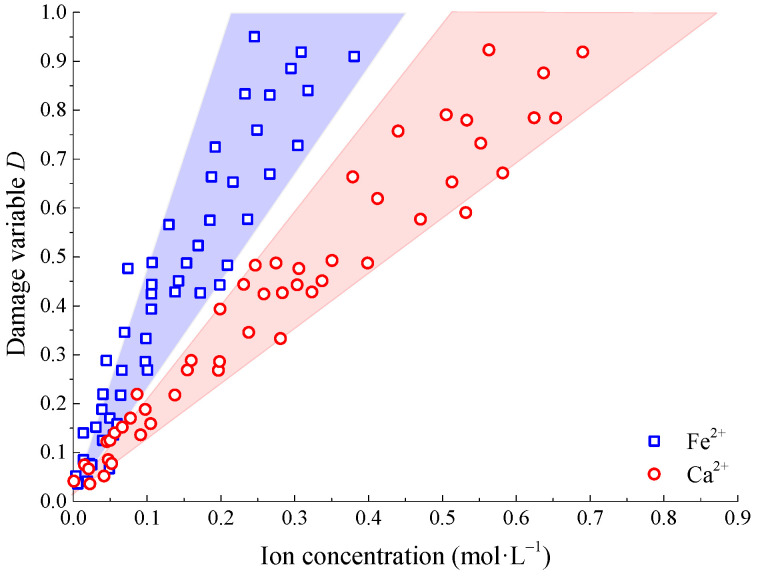
Relationship between damage variables and ion concentration.

**Table 1 materials-15-01613-t001:** Test scheme.

Chemical Solution	Confining Pressure *σ*_3_ (MPa)	Concentration *α* (mol·L^−1^)	pH
Na_2_SO_4_	5/10/15/20	0.02	2/4/6/7/8/10/12
		0.02/0.04/0.06/0.08/0.1	2

**Table 2 materials-15-01613-t002:** Mechanical parameters of Na_2_SO_4_ corroded sandstone (*σ*_3_ = 10 MPa).

Na_2_SO_4_		Peak Strength *σ*_pk_ (MPa)	Peak Strain *ε*_pk_ (%)	Elasticity Modulus *E* (GPa)	Poisson’s Ratio *µ*
pH	2	115.71	0.7351	16.07	0.371
	4	125.73	0.7746	16.71	0.342
	6	138.96	0.8542	17.42	0.314
	7	146.83	0.8654	17.98	0.293
	8	144.81	0.8391	17.73	0.302
	10	135.38	0.7944	17.05	0.327
	12	124.57	0.7532	16.49	0.346
*α* (mol·L^−1^)	0.02	115.71	0.7351	16.07	0.371
	0.04	112.14	0.7264	15.87	0.378
	0.06	101.62	0.7073	15.44	0.386
	0.08	97.04	0.6648	15.03	0.392
	0.10	86.54	0.6197	14.37	0.415

**Table 3 materials-15-01613-t003:** Characteristic stress of Na_2_SO_4_ corroded sandstone (*σ*_3_ = 10 MPa).

Na_2_SO_4_		*σ*_cc_ (MPa)	*σ*_ci_ (MPa)	*σ*_cd_ (MPa)	*σ*_pk_ (MPa)	*σ*_cc_/*σ*_pk_	*σ*_ci_/*σ*_pk_	*σ*_cd_/*σ*_pk_	*σ*_ci_/*σ*_cd_
pH	2	36.04	58.09	81.50	115.71	0.3115	0.5020	0.7043	0.3115
	4	38.26	63.89	91.13	125.73	0.3043	0.5082	0.7248	0.3043
	6	42.55	72.11	101.08	138.96	0.3062	0.5189	0.7274	0.3062
	7	44.58	74.69	107.19	146.83	0.3036	0.5087	0.7300	0.3036
	8	43.59	73.85	105.54	144.81	0.3010	0.5100	0.7288	0.3010
	10	41.35	68.57	98.50	135.38	0.3054	0.5065	0.7276	0.3054
	12	37.89	62.65	90.48	124.57	0.3042	0.5029	0.7263	0.3042
*α* (mol·L^−1^)	0.02	36.04	58.09	81.50	115.71	0.3115	0.5020	0.7043	0.3115
	0.04	35.47	56.11	80.56	112.14	0.3214	0.5004	0.7451	0.3214
	0.06	34.36	53.94	79.66	101.62	0.3430	0.5308	0.8036	0.3430
	0.08	33.07	48.46	74.83	97.04	0.3408	0.4994	0.7711	0.3408
	0.10	30.84	44.31	67.72	86.54	0.3564	0.5120	0.7825	0.3564

**Table 4 materials-15-01613-t004:** Characteristic stress of Na_2_SO_4_ corroded sandstone (*σ*_3_ = 10 MPa).

Na_2_SO_4_		*U* (kJ·m^−3^)	*U*^e^ (kJ·m^−3^)	*U*^d^ (kJ·m^−3^)	*U*^e^/*U*	*U* ^d^ */U*
pH	2	51.94	43.63	8.31	0.84	0.16
	4	58.69	51.06	7.63	0.87	0.13
	6	65.82	59.22	6.6	0.9	0.10
	7	80.51	74.06	6.45	0.92	0.08
	8	74.56	67.84	6.72	0.91	0.09
	10	64.91	57.13	7.78	0.88	0.12
	12	62.95	53.26	8.69	0.86	0.14
*α* (mol·L^−1^)	0.02	51.94	43.63	8.31	0.84	0.16
	0.04	44.70	36.65	8.05	0.82	0.18
	0.06	43.67	35.81	7.86	0.82	0.18
	0.08	37.74	30.57	7.17	0.81	0.19
	0.10	30.10	23.78	6.32	0.79	0.21

**Table 5 materials-15-01613-t005:** Parameter calculation results (*σ*_3_ = 10 MPa).

Concentration *α* (mol·L^−1^)	Parameters			
	*λ*	*β*	*B*	*n*
0.02	0.5381	−4.5465	1.8584	0.0106
0.04	0.6647	−4.7456	1.5044	0.0087
0.06	0.7299	−5.0459	1.3701	0.0064
0.08	0.7443	−5.1779	1.3435	0.0056
0.10	0.7655	−5.2211	1.3063	0.0054

## Data Availability

The data presented in this study are available on request from the corresponding author. The data are not publicly available due to privacy.

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
