# Peer review of "Study on Mechanical Behavior and Energy Mechanism of Sandstone under Chemical Corrosion"

_materials, 2022, doi:10.3390/ma15041613_

Round 1

Reviewer 1 Report

Dear Author,

the paper entitled Study on mechanical behavior and energy mechanism of sand-
stone under chemical corrosion” deals with the study of the chemical corrosion of sandstone, performing parametric studies on different variables. The laboratory experiments seems  to be conduce ed in a proper way as well as the analysis of the results. I personally recommend to add in the introduction section also the results/findings of the previous work, not limiting the section to a list of references. Also the term energy analysis should be briefly explained in the introduction. I suggest to stress morte the findings in the conclusions as well as in the abstract

Author Response

Reply for Reviewer 1,

First of all, thank you very much for taking time out of your busy schedule to review this paper and put forward valuable comments. The specific reply are as follows:

1) Reply: According to the comments, the introduction part has been modified, adding the summary of previous research results, and pointing out the shortcomings of previous research work combined with the summary and conclusion part.

2) Reply: According to the comments, the energy mechanism analysis is explained.

All changes are shown in red in the revised manuscript.

Thank you again for your valuable comments on this article.

Reviewer 2 Report

The paper explained the outcomes from the study on mechanical behaviour and energy mechanism of sandstone under chemical corrosion. The manuscript is well organized however needs the following improvement before accepting to the journal:

1) Section 2: please specify the standard used for triaxial tests.

2) Section 3.3 title repeated 2 times. needs rearrangement.

3) The results are well demonstrated however no deep discussion is provided. As a scientific paper, it's highly recommended to add an additional section or modify the result section to provide a detailed discussion.

4) Sandstone itself is usually subjected to weathering and it may affect the corrosion process, so, the authors should indicate this phenomenon within the discussion.

5) Creep and cracking also are the other significant properties of the Sandstone, so the authors should pay more attention to the achieved result from another viewpoint.

6) English needs to be checked by the technical native English speaker.

Author Response

Reply for Reviewer 2,

First of all, thank you very much for taking time out of your busy schedule to review this paper and put forward valuable comments. The specific reply are as follows:

1) Reply: The surrounding rock of underground engineering is usually in a complex stress environment. In order to simplify calculation and facilitate understanding, the surrounding rock of underground engineering is usually divided into two types, including axial stress (gravity stress) and confining pressure (in-situ stress). The mechanical properties of rock are different at different depths. Conventional indoor triaxial test is an international universal rock mechanics test method. The method applies axial pressure and confining pressure to standard cylindrical specimens by hydraulic oil. The confining pressure is adjusted by the control system to approximate the stress environment of rock at a certain depth underground, and then the mechanical properties of rock at this position are analyzed.

2) Reply: According to the comments, the title of section 3.4 is corrected - Analysis of energy mechanism

3) Reply: According to the comments, a discussion part has been added in this paper, and the mechanism of chemical corrosion of sandstone has been analyzed in detail.

4) Reply: According to the comments, the interaction between weathering and chemical corrosion has been pointed out in the discussion section.

5) Reply: According to the comments, creep and cracking analysis of chemically corroded sandstone is added in the discussion section.

6) Reply: According to the comments, this paper has been checked by native English speakers and some sentences have been modified.

All changes are shown in red in the revised manuscript.

Thank you again for your valuable comments on this article.